# Facilitation of molecular motion to develop turn-on photoacoustic bioprobe for detecting nitric oxide in encephalitis

Ji Qi[1,6], Leyan Feng[2,6], Xiaoyan Zhang[3], Haoke Zhang[1], Liwen Huang[2], Yutong Zhou[2], Zheng Zhao[1], Xingchen Duan[3], Fei Xu[2], Ryan T. K. Kwok [1,4], Jacky W. Y. Lam [1,4], Dan Ding[3], Xue Xue[2✉] & Ben Zhong Tang [1,4,5✉]

Nitric oxide (NO) is an important signaling molecule overexpressed in many diseases, thus the development of NO-activatable probes is of vital significance for monitoring related diseases. However, sensitive photoacoustic (PA) probes for detecting NO-associated complicated diseases (e.g., encephalitis), have yet to be developed. Herein, we report a NO-activated PA probe for in vivo detection of encephalitis by tuning the molecular geometry and energy transformation processes. A strong donor-acceptor structure with increased conjugation can be obtained after NO treatment, along with the active intramolecular motion, significantly boosting "turn-on" near-infrared PA property. The molecular probe exhibits high specificity and sensitivity towards NO over interfering reactive species. The probe is capable of detecting and differentiating encephalitis in different severities with high spatiotemporal resolution. This work will inspire more insights into the development of high-performing activatable PA probes for advanced diagnosis by making full use of intramolecular motion and energy transformation processes.

[1] Department of Chemistry, The Hong Kong Branch of Chinese National Engineering Research Center for Tissue Restoration and Reconstruction, Institute for Advanced Study, Department of Chemical and Biological Engineering and Institute of Molecular Functional Materials, The Hong Kong University of Science and Technology, Kowloon, Hong Kong, China. [2] State Key Laboratory of Medicinal Chemical Biology, College of Pharmacy, Nankai University, Haihe Education Park, Tianjin, China. [3] State Key Laboratory of Medicinal Chemical Biology, Key Laboratory of Bioactive Materials, Ministry of Education, and College of Life Sciences, Nankai University, Tianjin, China. [4] HKUST-Shenzhen Research Institute, Nanshan, Shenzhen, China. [5] NSFC Centre for Luminescence from Molecular Aggregates, SCUT-HKUST Joint Research Institute, State Key Laboratory of Luminescent Materials and Devices, South China University of Technology, Guangzhou, China. [6] These authors contributed equally: Ji Qi, Leyan Feng. ✉email: xuexue@nankai.edu.cn; tangbenz@ust.hk

As the basic nature of matter, molecular motion plays an important role in determining many fundamental chemical and physical processes[1,2]. Continued interest in the field of molecular motion has encouraged controlling and using the transduction of molecular motion energy, for example, to suppress or promote the related energy as needed, which would significantly benefit real applications[3,4]. For instance, the intramolecular motions (e.g., rotation, vibration, and twist) of a chromophore in the excited state are directly in association with energy transformation processes, and our previous researches about aggregation-induced emission (AIE) suggest that manipulation of the excited-state intramolecular motion could tune the fluorescence property in a highly efficient way[5–7]. Active intramolecular motion can promote the release of excitation energy via nonradiative decay, a process which determines how much absorbed light can be converted to heat and is closely linked to several important biological techniques such as photoacoustic (PA) imaging[8,9].

PA imaging is an emerging biomedical imaging technique that relies on ultrasound signals generated by thermoelastic expansions of optically-excited tissue or contrast agents[10,11]. By availing the benefits of optical resolution and acoustic depth of penetration, PA technique enables deep tissue imaging capacity with high spatial resolution and real-time monitoring, rendering great promise for clinical translation[12,13]. Organic PA imaging agents based on small molecules and polymers have captured intense attention as they possess intrinsic merits such as well-defined structure, large-scale production, ease-of-modification, and good biocompatibility[14,15]. PA molecular imaging based on these near-infrared (NIR) contrast agents has been explored for cancer detection, staging, and treatment guidance[16–19]. However, it remains a profound challenge to develop high-performing PA probe for accurate, in vivo diagnosis of complicated and deeply located diseases, like brain inflammation. Thus, it is critically important to exploit organic PA probes with high light-acoustic conversion capacity and high signal-to-background ratio (SBR) to achieve advanced biomedical diagnosis. The PA effect is closely linked to the photophysical transition process, which could be boosted by tuning the molecular structure and making full use of

the absorbed photoenergy[20,21]. For example, Pu et al. pioneered the intraparticle photoinduced electron transfer strategy to quench fluorescence and enhance nonradiative deactivation of semiconducting polymers by doping a fullerene derivative as the acceptor material into nanoparticles (NPs)[22]. Nevertheless, the introduction of a second component increases the complexity of the system. On the other hand, the simple structure with improved PA generation property would be more practically useful and merit exploration. As mentioned above, the active intramolecular motion of twisted structures can theoretically promote efficient nonradiative decay to release the excitation energy as heat, which represents a useful strategy to boost the performance of PA imaging agents. However, most molecular rotors are only active in solution, and exhibit limited intramolecular motion in aggregate forms such as NPs, which are more useful for biological applications[23–25]. Another issue for current PA imaging is that most reported PA probes are in "always-on" state and lack specificity. PA signals from surrounding normal tissue frequently interfere with signal at the site of interest, leading to false diagnostic outcomes[26,27]. One solution for this problem is the exploration of activatable PA contrast agents with "off-on" signal in response to a specific biomarker, which can substantially improve the SBR and real-time imaging capability in vivo[28–30].

Nitric oxide (NO) is an extremely important signaling molecule that plays key roles in regulating various physiological and pathological processes[31,32]. Studies have shown that NO is overexpressed in many diseases including endothelial dysfunction, cancer, neurodegenerative diseases, and cerebral infection[33–35]. For instance, the NO concentration in encephalitis site is usually far higher than normal brain tissue. More importantly, NO level could also reflect encephalitis in different stages[36]. Therefore, the detection of NO and related diseases is of critical significance for understanding disease severity and progression. Current approaches for detecting NO are inadequate for in vivo applications, especially real-time noninvasive monitoring of brain diseases. The most widely used method for analyzing NO is the colorimetric Griess assay, which requires acidic conditions and thus has limited in vivo applications[37,38]. Fluorescence imaging

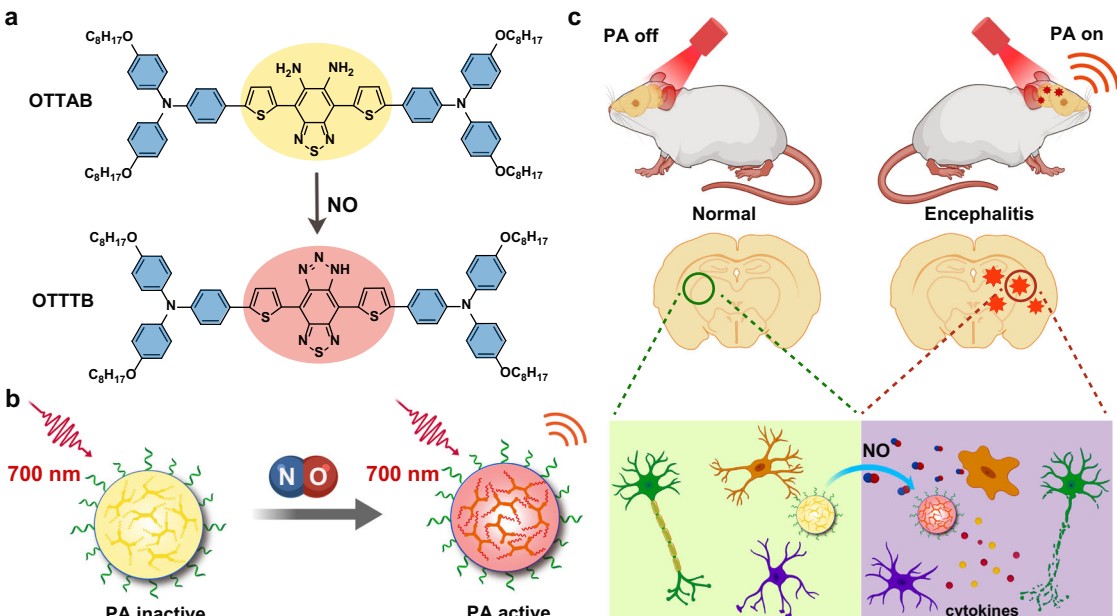

**Fig. 1 Turn-on PA probe for NO detection in vivo. a** Changes of chemical structures and **b** NPs property after NO treatment. **c** Schematic illustration of the in vivo detection of encephalitis with the turn-on PA probe. The illustrations were created with the help of BioRender.com.

has been developed as a popular method for probing NO with high sensitivity, yet it faces the drawback of shallow penetration depth, which restricts its use to the cellular level and skin surfaces[39–41]. Clinically used methods such as magnetic resonance imaging could realize nearly unlimited penetration depth. However, their low sensitivity and spatial resolution reduce the detection efficiency[42]. Owning to its intrinsic advantages, PA imaging represents a promising method to realize real-time detection of NO with high spatiotemporal resolution in vivo. However, it is challenging to develop highly specific and sensitive NO-activated PA nanoprobes to enable precise and in vivo diagnosis of NO-related disease in deep tissue, such as encephalitis.

In this work, we develop a sensitive "turn-on" PA probe with maximal energy transformation for in vivo detection of NO in encephalitis. We design and synthesize a molecular probe that can react with NO (Fig. 1a), which results in a relatively planar structure with strong intramolecular donor–acceptor (D–A) interaction, and thus a new absorption band in NIR region. Moreover, the molecule becomes highly twisted in the excited state, which further facilitates the active intramolecular motion and thus boosts PA conversion. As a result, by combining the planarization in ground state and twistification in excited state, a turn-on NO probe with enhancive PA signal is obtained (Fig. 1b). The probe not only exhibits good selectivity and quantitation toward NO, but also possesses high sensitivity, which enable excellent in vitro PA signal output. The probe has also been employed for noninvasive in vivo PA imaging of NO in encephalitis and differentiating its severity (Fig. 1c), greatly expanding the biomedical applications of active molecular motion as well as the applicability of PA imaging in advanced disease diagnosis and precision medicine.

## Results

**Design and synthesis of NO probe.** As shown in Fig. 1a, a molecular probe with reaction-tunable D–A interaction and conformation is designed and synthesized. The octyloxy-substituted triphenylamine (OT) is utilized as the donor unit for the strong electron-donating property and increased solubility. The planar thiophene (T) ring is function as both donor and π-bridge unit, which can further enhance the electron-donating ability and facilitate the intramolecular charge transfer (ICT). The diamine-substituted benzothiadiazole (AB) is selected as the reaction-tunable acceptor core because the *o*-phenylenediamino group could react with NO to afford a triazole product[43,44]. As a consequence, a more electron-deficient 5*H*-[1,2,3]triazolo [4′,5′:4,5]benzo[1,2-*c*][1,2,5]thiadiazole (TB) structure is formed, which enables much stronger D–A interaction and ICT effect. The molecular geometries are also expected to change after NO treatment as the steric hindrance between thiophene and the acceptor core differs. Moreover, the long aliphatic chains are designed to retain some flexible space between the conjugated backbones, which would be favourable for intramolecular motions in aggregate state (e.g., NPs)[8,45]. And the twisted phenyl rings of triphenylamine would also benefit the intramolecular motion. Herein, a D-(NO activatable A)-D type probe (OTTAB) was designed and synthesized, and the synthetic route is depicted in Supplementary Fig. 1. The intermediates and final product have been characterized by nuclear magnetic resonance (NMR) and high-resolution mass spectrum (HRMS) (Supplementary Figs. 2–15), and the product (OTTTB) after NO treatment has also been verified by HRMS (Supplementary Fig. 16).

**Photophysical properties before and after NO treatment.** The absorption spectra of OTTAB and OTTTB in THF are displayed

in Fig. 2a. Interestingly, the long-wavelength absorption peak shifts from 459 to 686 nm, allowing pronounced turn-on NIR PA signature. It is noted that OTTTB exhibits a high absorption coefficient of $3.16 \times 10^4 \, M^{-1} \, cm^{-1}$, which is due to the planar conjugated core structure and will ensure strong PA generation property. The NIR absorption band of OTTTB can be assigned to the efficient ICT effect of strong D–A interaction[46,47]. To investigate the intramolecular motion in aggregate states, water was gradually added into the THF solution, and the PL spectra were recorded (Fig. 2b, c). The PL intensity of OTTAB decreases at first and then increases gradually with water fraction, representing a kind of AIE signature. On the contrary, only very weak PL signal can be observed for OTTTB in high water fractions (Fig. 2d), which reveals that intramolecular motion is intense in aggregate states and the excited-state energy is dissipated through non-radiative decay[48]. Moreover, the maximal PL of OTTTB red shifts gradually from 851 to 933 nm as the water fraction increases (Supplementary Fig. 17), demonstrating typical twisted intramolecular charge transfer (TICT) effect in high polarity environment. Therefore, dark TICT state is formed in aggregates, in which the active intramolecular motion would facilitate the nonradiative relaxation to generate PA signal[49]. The different photophysical behaviors of OTTAB and OTTTB may be due to the structure change.

Density function theory (DFT) calculation was conducted to study the molecular geometries. As shown in Fig. 3a, the dihedral angles between the acceptor core and thiophene rings are greatly decreased from 38.5° and 36.7° for OTTAB to 2.8° and 15.9° for OTTTB, as the steric hindrance between amino groups and thiophene spacer becomes much higher. The planar structure obtained after NO treatment enables better conjugation and more efficient ICT, and thus new NIR absorption. As a result, the pronounced alteration in molecular structure of NO treatment leads to distinct photophysical energy conversion processes, in which the maximal nonradiative process of OTTTB would boost the turn-on PA property. As illustrated in Fig. 3b, the electronic bandgaps of OTTAB and OTTTB are 2.57 and 1.67 eV, respectively. The highest occupied molecular orbital (HOMO) energy level increases slightly (from −4.43 to −4.40 eV), while the lowest unoccupied molecular orbital (LUMO) energy level decreases a lot (from −1.86 to −2.73 eV). The HOMO and LUMO energy levels of D–A type compound are usually related to the electronic properties of donor and acceptor moieties, respectively. Consequently, this result confirms that the electron-withdrawing property of the acceptor core is activated by NO treatment, and strong D–A interaction is realized.

**Preparation and characterization of NPs.** To endow the hydrophobic compound with good water solubility and biocompatibility, OTTAB was encapsulated into NPs in the assistant of 1,2-distearoyl-*sn*-glycero-3-phosphoethanolamine-*N*-[methoxy-(polyethylene glycol)-2000] (DSPE-PEG$_{2000}$) with nanoprecipitation method (Fig. 4a). The hydrophobic molecule is self-assembly in the core, and the amphiphilic lipid-PEG polymer functions as the surface layer. It is noted that the long alkyl chain in DSPE-PEG could tangle with the octyl substitutes in OTTAB, which endues the encapsulated NPs with some flexible domains beneficial for free movement. Accordingly, the NPs' PA effect is expected to promote by the active intramolecular motions. Transmission electron microscopy (TEM) measurement (Fig. 4b) suggests that the obtained organic NPs are a kind of sphere structure, and dynamic light scattering (DLS) result gives an average diameter of 142 nm. Due to the hydrophobic nature of the organic probe, the encapsulated NPs are very stable in aqueous media, and the diameter nearly does not change after storage

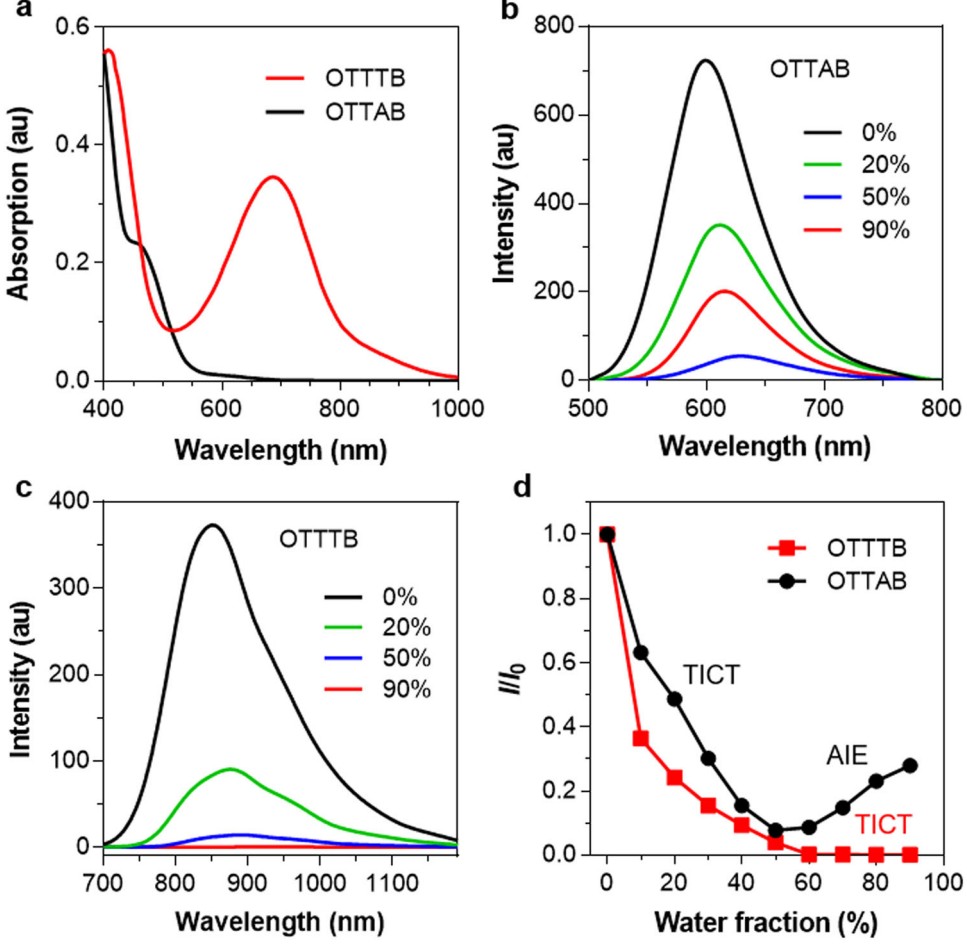

**Fig. 2 Photophysical properties before and after NO treatment. a** Absorption spectra of OTTAB and OTTTB in THF ($10^{-5}$ M). PL spectra of **b** OTTAB and **c** OTTTB ($10^{-5}$ M) in THF/water mixture with various water fractions ($f_w$) as indicated (The excitation wavelengths of OTTAB and OTTTB are 460 and 680 nm, respectively.) **d** Plot of the PL peak intensity of OTTAB and OTTTB ($10^{-5}$ M) versus water fractions in THF/water mixture. $I_0$ and $I$ are the PL peak intensities in pure THF ($f_w = 0$) and THF/water mixtures with specific $f_w$s, respectively.

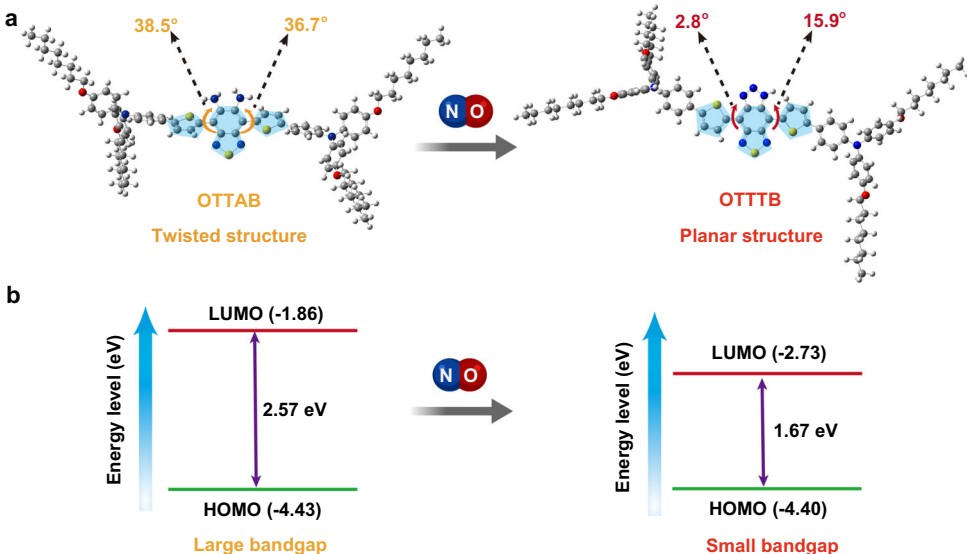

**Fig. 3 Changes of molecular conformation and energy level after NO treatment. a** The optimized molecular geometries and **b** corresponding changes in the electronic energy levels of OTTAB before and after NO treatment.

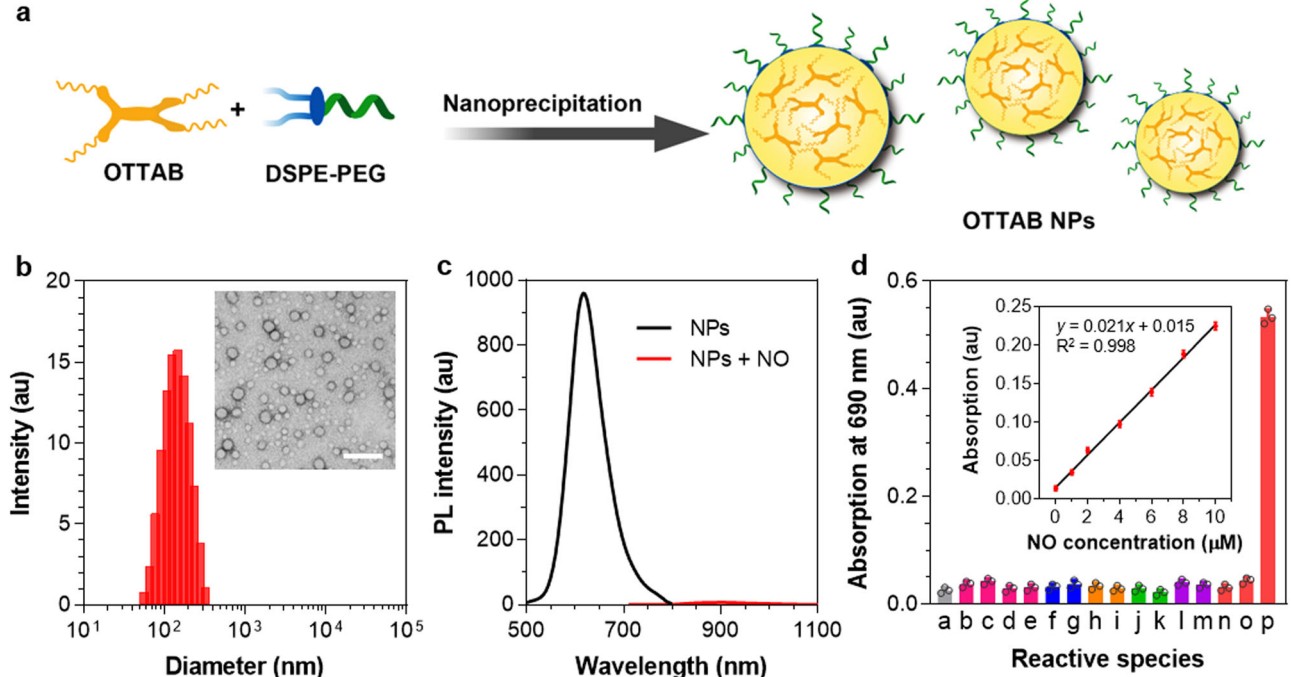

**Fig. 4 Characterizations and reactivity of the NPs. a** Schematic illustration showing the nanoprecipitation method. **b** Representative DLS and TEM results of the NPs. Scale bar: 500 nm. **c** PL spectra of OTTAB NPs ($10^{-5}$ M) before and after NO treatment (The excitation wavelengths before and after NO treatment are 460 and 680 nm, respectively.) **d** Changes of the absorption intensity of OTTAB NPs ($2 \times 10^{-5}$ M) at 690 nm with the treatment of different species. a: no treatment, b: $Ca^{2+}$, c: $Fe^{2+}$, d: $Fe^{3+}$, e: $Cu^{2+}$, f: GSH, g: Hcy, h: $H_2O_2$, i: •OH, j: $O_2^-$, k: $ClO^-$, l: $SO_3^{2-}$, m: $H_2S$, n: $NO_3^-$, o: $ONOO^-$, p: NO. The concentrations of all the species were 1 mM. Inset shows the relationship between the absorption intensity of OTTAB NPs ($2 \times 10^{-5}$ M) at 690 nm and the concentration of NO. Data are presented as mean ± s.d. ($n = 3$ independent experiments).

in ambient condition for two weeks (Supplementary Fig. 18). Moreover, the NPs also exhibit good stability in different conditions such as Dulbecco's modified Eagle's medium (DMEM) and acidic/basic environments (Supplementary Fig. 19), which manifest potential in vivo applications.

The absorption spectra of the NPs before and after NO treatment are presented in Supplementary Fig. 20, and the reaction kinetics of OTTAB NPs toward NO is displayed in Supplementary Fig. 21. Interestingly, a new absorption band with maximum at about 690 nm appears after reacting with NO, matching well with the absorption spectrum of OTTTB (Fig. 2a). The NPs possess strong absorption in the spectral region of 600–800 nm, which locates within the excitation range of commercially available PA instrument (680–950 nm). Meanwhile, the color of NPs solution apparently changes from yellow to green (Inset of Supplementary Fig. 19). The large bathochromic shift of about 230 nm in absorption spectra is sufficient for eliminating the signal background from pristine probe. We further studied the PL property as it is a competitive channel to nonradiative PA generation pathway. OTTAB NPs exhibit strong emission, while only very weak PL signal can be observed after NO treatment, indicating that the radiative process is greatly suppressed due to the more planar structure and reduced bandgap (Fig. 4c), being favourable for nonradiative decay and PA generation. It is noted that the maximal PL of NO-treated OTTAB NPs is at 902 nm, which suggests the formation of dark TICT state, and strong intramolecular motions promoting nonradiative decay within the NPs. To verify the specificity of the probe for sensing NO, we next determined the detection selectivity. A lot of possible reactive species including metal ions, amino acids, and reactive oxygen/nitrogen/sulfur species (ROS/RNS/RSS) have been screened (Fig. 4d). Of note, the probe is inert to other ROS, RNS and RSS, such as $H_2O_2$, •OH, $O_2^-$, $ClO^-$, $SO_3^{2-}$, $H_2S$, $NO_3^-$, and

$ONOO^-$ [50]. Moreover, the probe exhibits neither response to metal ions, nor GSH/Cys. By comparison, only NO treatment could generate strong NIR absorption, implying the high selectivity of OTTAB NPs. The relationship between the absorption intensity at 690 nm and NO concentration was further investigated (Inset of Fig. 4d and Supplementary Fig. 22), which demonstrates very good linear behavior ($R^2 = 0.998$). In addition, the detection limit ($3\sigma$/slope) of OTTAB NPs for NO is calculated to be 377 nM. The dose-dependent responsivity toward NO suggests that the probe is suitable for quantitative analysis.

**In vitro PA response of the nanoprobe.** After confirming the photophysical properties, we next studied the PA response of this probe. PA spectrum of the NO-treated OTTAB NPs is displayed in Fig. 5a, which is in agreement with the absorption profile, and reveals that the PA signal indeed comes from the NIR absorption of OTTTB. The PA stability was then evaluated by scanning the NO-treated NPs in a phantom with $1.2 \times 10^4$ of laser pulses at 700 nm (17.5 mJ cm$^{-2}$ laser and 10 Hz pulse repetition rate). Noteworthy, the probe exhibits pretty stable PA signal as there is negligible change in PA amplitude after exposure to pulsed laser, much better than that of clinically used indocyanine green (Fig. 5b). This verifies the superb stability and suitability of the probe for long-term PA imaging. The PA signal outcome was further studied by recording the PA intensity of NO-treated OTTAB NPs in different concentrations, and linear correlation between the PA amplitude and probe concentration is observed (Fig. 5c). The NO detection capacity of the probe was examined by measuring the PA response with the treatment of different concentrations of NO. As displayed in Fig. 5d, a good linear relationship is observed in relatively low NO concentrations (<2.5 μM), while the reaction tends to become saturation when further increasing NO concentration. This reveals that the probe

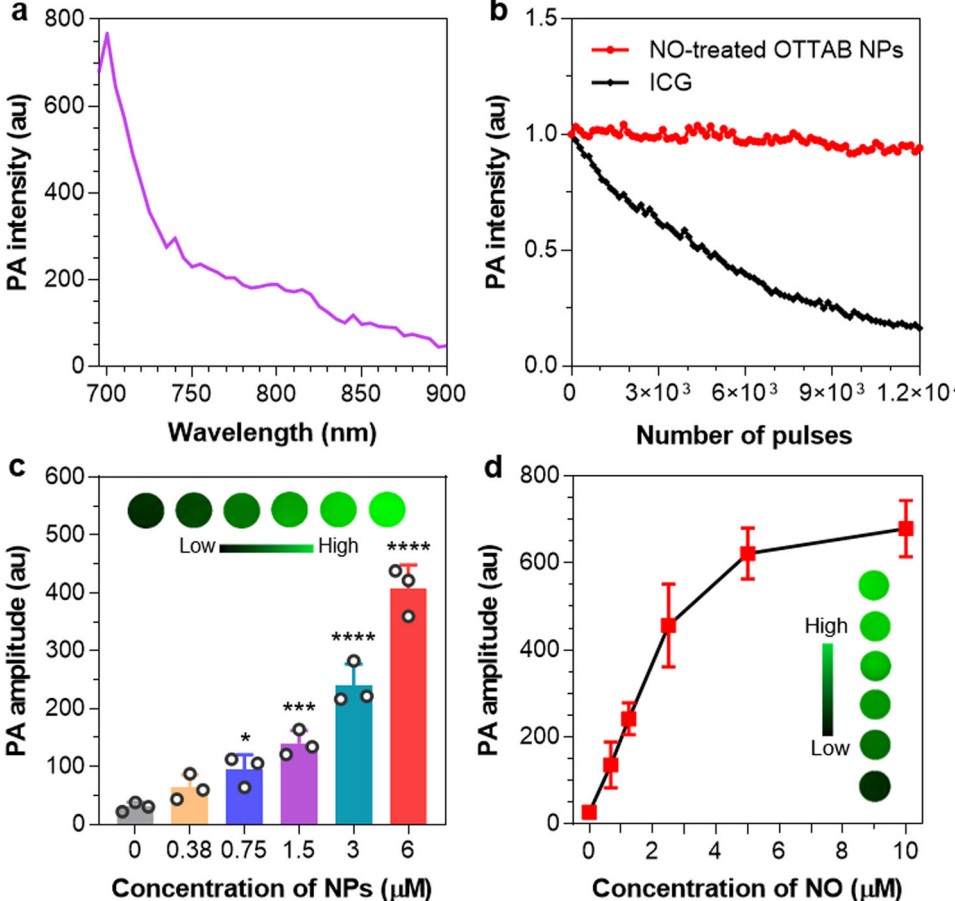

**Fig. 5 In vitro detection of NO. a** PA spectrum of the NO-treated OTTAB NPs (10 µM). **b** Plots of the PA intensity of NO-treated OTTAB NPs ($\lambda_{ex} = 690$ nm) and indocyanine green (ICG, $\lambda_{ex} = 790$ nm) in a phantom (10 µM) against number of laser pulses (17.5 mJ cm$^{-2}$ laser and 10 Hz pulse repetition rate). **c** PA amplitudes and the corresponding representative PA images of the NO-treated OTTAB NPs as a function of concentration. Data are presented as mean ± s.d. ($n = 3$ independent experiments). *$p = 0.017$, ***$p = 0.00049$, ****$p < 0.0001$ (3 vs. 0: $p = 1 \times 10^{-6}$, 6 vs. 0: $p = 1.5 \times 10^{-9}$) compared to 0 µM of NPs using one-way ANOVA for multiple comparisons. **d** PA amplitudes and the corresponding representative PA images of OTTAB NPs (10 µM) after the treatment with different concentrations of NO. Data are presented as mean ± s.d. ($n = 3$ independent experiments).

holds great potential for in situ PA detection of NO-related diseases in different stages. In vitro cellular viability experiments show that more than 90% of cells maintain alive after treating with a high concentration of the nanoprobe (50 µM) (Supplementary Fig. 23), revealing good cell compatibility. Taken together, these results reveal that OTTAB NPs is a superb probe for NO sensing, encouraging more exploration for detecting related severe diseases such as encephalitis in situ.

**In vivo PA imaging of encephalitis**. Encephalitis, or inflammation of brain tissue, is a kind of serious cerebral disease that can cause headache, fever, mental confusion, seizures, and even death[51,52]. Accurate and high-resolution diagnosis of encephalitis in real time greatly benefits effective treatment and successful recovery from encephalitis. It has been reported that brain cells such as microglia, astrocytes, neurons, and endothelial cells would be activated during inflammatory pathologies, and upregulate the expression of nitric oxide synthases (NOS)[53,54]. NOS can catalyze the oxidation of L-arginine to L-citrulline and generate a large amount of NO, which is considered as an important inflammatory mediator that correlates with severity. However, noninvasive in vivo detection of NO-associated encephalitis remains a challenging task due to the lack of highly sensitive and deep-penetrating imaging technology. We thus investigated the

feasibility of our PA nanoprobe for detecting NO in encephalitis in living mice. The murine encephalitis model was built by intracerebroventricular administration of lipopolysaccharide (LPS) into mouse brain. LPS, an endotoxin from the outer membrane of bacteria, is known as a potent trigger of inflammation[55,56]. It has been demonstrated that LPS induces astrocyte and microglia activation, as well as overexpression of NOS and other pro-inflammatory cytokines in brain[57].

Herein, the right lateral ventricle of mouse brain was pretreated with LPS, whereas the contralateral hemisphere of brain was treated with saline to serve as a control. OTTAB NPs solution (1 mg mL$^{-1}$, 2 µL) was intracerebroventricularly injected to investigate the feasibility for detecting NO-related encephalitis in vivo. PA imaging of mouse brain with intact skull was performed with 700 nm excitation. A representative PA image of the whole mouse brain clearly illuminates that only the inflamed site shows light-up PA signal after injecting the nanoprobe (Supplementary Fig. 24), suggesting pronounced turn-on PA property of our probe. The PA imaging results are shown in Fig. 6a, and corresponding PA intensities of brains at different time points post administration are depicted in Fig. 6b. Immediately after the NPs administration, both the LPS-injected and saline-injected intervals exhibit negligible PA signal, which indicates that LPS or saline treatment does not generate obvious signal interference.

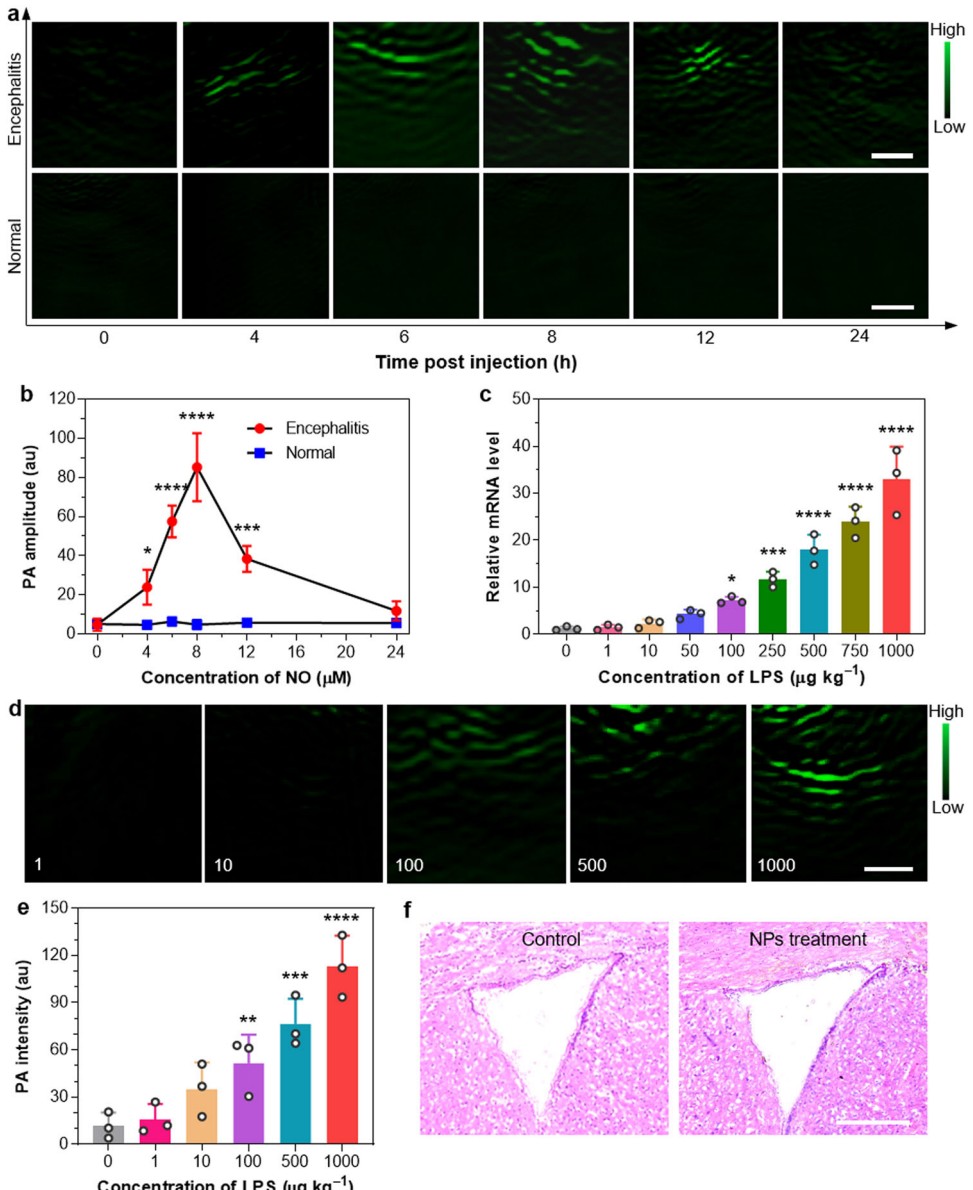

**Fig. 6 In vivo PA imaging of encephalitis in different severities. a** Representative in vivo noninvasive PA images and **b** the corresponding PA intensity of saline-treated brain (normal) and LPS-induced encephalitis after infusing OTTAB NPs at different time points as indicated. Scale bars: 1 mm. **c** Relative inducible nitric oxide synthase (iNOS) mRNA level at the disease site versus the concentration of injected LPS. **d** Representative in vivo noninvasive PA images and **e** corresponding PA intensity of the mouse brains treated with different concentrations ($\mu g\,kg^{-1}$) of LPS as indicated after administrating OTTAB NPs for 8 h. Scale bar: 1 mm. Data are presented as mean ± s.d. ($n = 3$ biologically independent mice per group) for **b**, **c**, **e**. **f** Histological H&E staining of the mouse brain without and with the treatment of OTTAB NPs. Scale bar: 250 μm. Brain tissues were harvested from three mice in each group, and the representative images from each group are shown. *$p < 0.05$ (4 vs. 0: $p = 0.028$ in **b**; 100 vs. 0: $p = 0.021$ in **c**), **$p < 0.01$ (100 vs. 0: $p = 0.0082$ in **e**), ***$p < 0.001$ (12 vs. 0: $p = 0.00087$ in **b**; 250 vs. 0: $p = 0.00032$ in **c**; 500 vs. 0: $p = 0.00025$ in **e**), ****$p < 0.0001$ (6 vs. 0: $p = 1.7 \times 10^{-5}$, 8 vs. 0: $p = 2.1 \times 10^{-7}$ in **b**; 500 vs. 0: $p = 1.3 \times 10^{-6}$, 750 vs. 0: $p = 1.7 \times 10^{-8}$, 1000 vs. 0: $p = 7.9 \times 10^{-11}$ in **c**; 1000 vs. 0: $p = 3.6 \times 10^{-6}$ in **e**) using one-way ANOVA for multiple comparisons.

Interestingly, PA intensity of the inflamed right ventricle increases rapidly over time, and reaches to maximum at 8 h. In contrast, PA signal of the saline-treated left ventricle nearly does not change throughout the experiment. Quantitative analysis further illuminates that PA intensities of the inflamed ventricles are about 5- and 17.7-fold higher than those of the control left ventricles at 4 and 8 h post administration, respectively, indicating that OTTAB NPs could sensitively monitor the endogenously generated NO within encephalitis in vivo. It is also noted that the PA imaging of encephalitis site exhibits a significantly high SBR of 15.7 at 8 h post-injection, which can be

attributed to the excellent turn-on PA responsivity of OTTAB NPs toward NO.

To further explore the utility of our PA nanoprobe for encephalitis evaluation, various amounts of LPS (1, 10, 100, 500, and 1000 $\mu g\,kg^{-1}$) were used to provoke inflammatory response in mice brain. To study the influence of different amounts of LPS treatment on encephalitis, the mice brains were collected to analyze the NOS level. NOS are not only an important indicator involved in inflammation response but also an enzyme that catalyzes the production of NO in encephalitis. Thus, the NOS level in brain could reflect the disease severity and NO generation

directly. NOS mainly include three kinds of isoforms, i.e., inducible NOS (iNOS), neuronal NOS (nNOS), and endothelial NOS (eNOS)[58,59]. In different types of encephalitis, the expression level of one or two or all three isoforms has been shown to increase significantly to induce the generation of NO. So we first evaluated the expression of them in the LPS-induced encephalitis model. The mRNA expression levels of eNOS, nNOS, and iNOS (respectively, represented by bEnd 3, N2A, and BV-2 cell lines) were analyzed by quantitative reverse transcriptase polymerase chain reaction (PCR). As displayed in Supplementary Figs. 25, 26, both in vitro and in vivo results indicate that the iNOS level is much higher than the other two analogs. Accordingly, the upregulated expression of iNOS is considered as the main contributor to NO production in this work. As presented in Fig. 6c, the iNOS mRNA levels of brain are elevated drastically with the doses of LPS used for treatment. The immunofluorescent staining analyses (Supplementary Fig. 27) on normal and inflamed brain tissues also reveal that the iNOS expression level in inflamed brain is far higher than that of normal one. We further measured the NO concentration at the encephalitis site using Griess assay (Supplementary Fig. 28), which has been widely used to detect the nitrite product formed by the spontaneous oxidation of NO under physiological condition. Interestingly, the NO concentrations are also found to increase with the doses of LPS used for stimulation, which is consistent with the expression of iNOS. Then in vivo PA imaging of mouse brain was conducted at 8 h post administration of OTTAB NPs to study the encephalitis severity in situ. Encouragingly, the PA intensities in brains consistently intensify with the injected LPS concentrations (Fig. 6d, e), and significantly stronger PA signals are observed from the brains pretreated with higher amounts of LPS, which are probably due to that higher LPS concentration induces more severe encephalitis. This result correlates well with the changes of iNOS expression and NO concentration, and the relationship of "encephalitis severity-iNOS level-NO concentration-PA intensity" has thus been established. OTTAB nanoprobe could be used for accurately detecting the NO-associated encephalitis in living mice and differentiating their severities, rendering great promise for understanding the disease progression and screening drugs. In addition, hematoxylin and eosin (H&E) staining of the OTTAB NPs-treated mouse brain (Fig. 6f) shows no obvious side effect, suggesting good in vivo biocompatibility.

The nanoprobe has been injected into mouse brain intracerebroventricularly to validate the efficacy of in vivo NO detection and related encephalitis imaging. The probe can also be developed for encephalitis imaging through intravenous administration by employing some strategies, such as modification of NPs with suitable ligands or certain cell membrane, to facilitate their penetration across the blood-brain barrier (BBB)[60,61]. For example, neutrophils (NEs) have been reported to possess an innate ability to penetrate BBB, which has been utilized to deliver NPs into brain[62,63]. In this work, OTTAB NPs were incubated with NEs to obtain OTTAB@NEs for enhanced brain delivery. The in vitro PA measurements (Supplementary Fig. 29) verify that OTTAB@NEs maintain excellent responsivity to NO. Further in vivo PA imaging results of the intravenously injected OTTAB@NEs reveal potent encephalitis imaging ability of the probe after intravenous administration (Supplementary Fig. 30).

## Discussion

In this work, we developed a sensitive NO probe with turn-on PA signature for noninvasive in vivo detection of NO in encephalitis. The molecular structure and photophysical transition processes of OTTAB probe change greatly after NO treatment, which

results in a strong NIR absorption band and turn-on PA signal output. The NO-activatable PA probe with maximal energy transformation enables a high signal-to-noise ratio. Moreover, the twistification in dark TICT state and the flexible alkyl substitutes significantly facilitate the intramolecular motions in NPs, thus the absorbed light can convert into acoustic signal maximally to significantly boost PA effect of the nanoprobe. The probe exhibits excellent sensitivity and selectivity toward NO over other interfering reactive species. In addition, the organic nanoprobe shows good biocompatibility, and no side effect was observed from both the in vitro and in vivo tests. The pronounced turn-on PA property motivates us to further evaluate the nanoprobe in encephalitis mice. Noninvasive in vivo PA imaging with the probe allows for detecting NO in encephalitis in a high-contrast manner. The probe is also capable of differentiating encephalitis of different severities, being beneficial for understanding the disease evolution processes and drug screening. This work reports a NO-activated probe for in vivo PA imaging of encephalitis in situ. The design strategy of fully utilizing the intramolecular motion in NPs would shed light on the exploitation of more sensitive bioprobes. This work will attract more insights into the development of highly efficient activatable PA probe for precise biomedical imaging, rendering great promise for real applications.

## Methods

**Characterizations**. $^1$H (400 MHz) and $^{13}$C (100 MHz) nuclear magnetic resonance (NMR) spectra were recorded on a Bruker AV 400 spectrometer using CDCl$_3$ as the solvent. High-resolution mass spectra (HRMS) were carried out on a GCT premier CAB048 mass spectrometer. The absorption spectra were measured using a Shimadzu 2550 UV-vis scanning spectrophotometer. The photoluminescence (PL) measurement was carried out on a Horiba Fluorolog-3 Spectrofluorometer. Dynamic light scattering (DLS) measurement was performed on a Malvern Zetasizer Nano ZS-90. Transmission electron microscope (TEM) images were captured by JEOL JEM-1200EX microscope with an accelerating voltage of 80 kV. Molecular geometry optimization was calculated using the DFT method with the Gaussian 09 program package (revision D. 01) at the level of B3LYP/6-31G*, and the Cartesian coordinates are presented in Supplementary Tables 1 and 2.

**Preparation of NPs**. 1 mg of OTTAB and 2 mg of DSPE-PEG$_{2000}$ were dissolved in 1 mL of tetrahydrofuran (THF) solution, which was poured into 9 mL of deionized water. Followed by sonication with a microtip probe sonicator (XL2000, Misonix Incorporated, NY) for 2 min. The residue THF solvent was evaporated by violent stirring the suspension in fume hood overnight, and colloidal solution was obtained and used directly. The NPs solution of various OTTAB concentrations was obtained by condensation or dilution.

**Cytotoxicity study**. 3-(4,5-Dimethyl-2-thiazolyl)-2,5-diphenyl-tetrazolium bromide (MTT) assay was used to evaluate the cytotoxicity of OTTAB NPs. Detroit 551 cells were harvested in a logarithmic growth phase and seeded in 96-well plates ($2 \times 10^4$ cells per well with 100 μL suspension) and grew to ~80% confluence. Then the culture medium was replaced with 100 μL of fresh culture medium containing OTTAB NPs with various concentrations (the concentrations based on OTTAB were 0, 2, 5, 10, 20, and 50 μM, respectively). After incubating for 24 h, the culture medium was removed and the wells were washed three times with PBS, and 100 μL of MTT dissolved in serum-free culture medium (0.5 mg mL$^{-1}$) was added into each well. After 4 h, the MTT solution was removed cautiously and 100 μL of dimethyl sulfoxide (DMSO) was added into the wells, followed by gently shaking for 10 min. Then, the absorbance of MTT at 490 nm was measured by a Bio-Rad 680 microplate reader to evaluate the cell viability.

**Cell culture and stimulation of lipopolysaccharide**. The mouse microglial BV-2 cell lines were grown in high glucose DMEM supplemented with 10% fetal bovine serum (FBS), 100 IU mL$^{-1}$ penicillin, and 10 μg mL$^{-1}$ streptomycin. The cells were maintained in a humidified incubator with 95% air and a 5% CO$_2$ atmosphere at 37 °C. The medium containing appropriate agents were replaced every other day. The BV-2 cells were treated with LPS (100 ng mL$^{-1}$), and incubated for 24 h. After BV-2 cells were treated with LPS, NO would be generated and then diffused into the cell medium. The medium was collected as the NO source for further in vitro experiments. The NO concentration in the medium was measured to be 22.2 μM. The solution with various NO concentrations was obtained after different-fold dilution of the stock medium.

**Determination of NO concentration**. The NO concentration was determined using Griess assay. According to previous papers, nitrite is the stable end product of nitric oxide radical, and widely used as the indicator of NO[64,65]. Briefly, 50 μL of LPS-treated BV-2 cell medium or mouse brain homogenate was incubated with 50 μL of Griess reagent (sulfanilamine in 1% $H_3PO_4$/0.1% $N$-(1-naphthyl)-ethylene-diamine dihydrochloride/1% $H_3PO_4$/distilled water, 1:1:1:1) at room temperature for 10 min. The absorbance at 540 nm was measured via a microplate reader (SpectraMax i3x software: SoftMax Pro 6 (version 6.5.1)). The standard curve was prepared with various known concentrations (ranging from 0.1 to 40 μM) of $NaNO_2$.

**Animals experiments**. All animal studies were conducted under the guidelines set by Tianjin Committee of Use and Care of Laboratory Animals, and the overall project protocols were approved by the Animal Ethics Committee of Nankai University.

**Encephalitis model**. The Kunming male mice (6–8 weeks) purchased from SPF (Beijing) Biotechnology Co., Ltd. were used for the animal experiments. The housing condition of mice is the specific pathogen-free (SPF) laboratory animal center with free access to food and water, the humidity keeps at 30–70% (20–26 °C), and keeps at 12 h of light and 12 h of dark cyclic condition. In short, the mice were first anesthetized with 3.5% isoflurane through intraperitoneal injection. Then the mice were fixed on a brain stereotaxic apparatus (Stoelting, 51500D, USA). The scalp was cut and a hole was drilled at certain position on the skull (1.0 mm lateral, 0.2 mm posterior from bregma). A 5 μL of Hamilton syringe with a needle was then inserted through the hole into the lateral ventricle at 2.5 mm below the horizontal plane of bregma. Different concentrations of LPS in saline and saline (2 μL) were infused at the right and left lateral ventricles, respectively.

**In vivo PA imaging**. For intracerebroventricular injection, OTTAB NPs (1 mg mL$^{-1}$, 2 μL) were infused at the lateral ventricle. Then, the scalp was sewed and the mice were anesthetized using 2% isoflurane in oxygen. In vivo PA imaging of the mouse brain was performed in particular time intervals. The PA imaging was carried out on a commercial small-animal opt-acoustic tomography system (MOST, inVision 256-TF, iThera Medical, Germany). A wavelength-tunable (680–950 nm) optical parametric oscillator (OPO) pumped by a Nd:YAG laser provides excitation pulses with a duration of 7 ns at a repetition rate of 10 Hz. The light from the fiber covers an area of ~4 cm$^2$ with a maximum incident pulse energy of ~70 mJ at 700 nm (100 mJ, 70% fiber coupling efficiency). This generates an optical fluence of 17.5 mJ cm$^{-2}$, which is well within the safe exposures according to the American National Standard for Safe Use of Lasers. And PA imaging was performed at 700 nm excitation. The images then were reconstructed using the model-based algorithm supplied within the View MSOT software suite (V3.6, iThera Medical). To demonstrate the application of the probe for intravenous administration, we utilized neutrophils (NEs) as carriers to deliver the nanoprobe for enhanced brain delivery. Briefly, mature NEs were isolated from murine bone marrow[61]. And $1 \times 10^5$ of NEs were then incubated with OTTAB NPs (20 μg mL$^{-1}$) for 2 h at 37 °C. After washing with ice-cold PBS thrice, the OTTAB@NEs suspension was obtained and used immediately for subsequent study. For intravenous injection, OTTAB@NEs were suspended in 150 μL of saline solution ($5 \times 10^6$ of NEs), and administrated into mice via tail vein. The in vivo PA imaging was then performed as described above.

**RNA extraction and quantitative reverse transcription PCR**. After perfusion of the heart with ice-cold PBS, the mouse brain was collected. The cortex then was isolated and grinded in 1 mL of TRIzol on ice, and RNA was extracted using TRIzol reagent following the protocols supplied by the manufacturer. One microgram of total RNA was reverse-transcribed into cDNA using PrimeScript RT Master Mix. Then, real-time polymerase chain reaction (PCR) was conducted using SYBR® GREEN following the manufacturer's instructions with the help of Real-Time PCR Detection Systems Bio-red CFX Maestro (version 1.0). The following primer sets were used: iNOS, forward, 5′-CGT AGC AAA CCA CCA AGT-3′; reverse, 5′-GGT ATG AGA TAG CAA ATC GG-3′; eNOS, forward, 5′-TTC CTG GAC ATC ACT TCC CC-3′; reverse, 5′-CTT CCA TTC TTC GTA GCG CC-3′; nNOS, forward, 5GGT CGC TTT GAG TAC CAG CCT-3′; reverse, 5′-GGT CGC TTT GAC TCT CTT GG-3′; GAPDH, forward, 5′-TTC TCA GCC CAA CAA TAC A-3′; reverse, 5′-CCT TGT GGT GAA GAG TGT-3′. The housekeeping gene GAPDH was used for normalization. qPCR was conducted in triplicate for each sample, and target mRNA levels were quantified using the $2^{-\Delta\Delta CT}$ method.

**Immunofluorescent staining**. The sliced tissues of mice brains on coverslip were washed three times with PBS, and blocked with 0.4% Triton X-100 in Goat Serum for 1 h at room temperature. After washing with PBS thrice, the sections were incubated with anti-iNOS antibody (D6B6S, Cell Signaling Technology, Inc.) (diluted in Goat Serum, 1:1000) overnight at 4 °C. Subsequently, the tissue sections were washed three times with PBS and incubated with the secondary antibody, fluorescein (FITC)-conjugated AffiniPure Goat anti-Rabbit IgG (H + L) (Jackson Immuno Research LABORATORIES, INC, dilution 1:200) for 1.5 h at room temperature in a shading box. The nuclei were stained with 4,6-diamidino-2-phenylindole (DAPI, Vector Laboratories, Inc). The images were captured using a fluorescence microscope (CKX41, OLYMPUS IMAGING).

**Hematoxylin and eosin (H&E) staining**. The in vivo biocompatibility of OTTAB NPs was evaluated with H&E staining. In short, the mice were first anesthetized with 3.5% isoflurane through intraperitoneal injection. Then OTTAB NPs in saline (1 mg mL$^{-1}$, 2 μL) or saline (2 μL) were intracerebroventricularly administrated as described in the in vivo PA imaging section. The mice were sacrificed 24 h after the injection. The brain tissues were harvested and fixed using 4% paraformaldehyde, and then embedded for frozen section. The H&E-stained slices were imaged by a digital microscope (Leica QWin).

**Data analysis**. Statistical comparisons were made by unpaired Student's $t$-test (between two groups) and one-way ANOVA (for multiple comparisons). $p$ value <0.05 was considered statistically significant. All statistical calculations were carried out with GraphPad Prism (version 8.0.2).

**Reporting summary**. Further information on research design is available in the Nature Research Reporting Summary linked to this article.

## Data availability

All the data supporting the findings in this study are available in the paper and Supplementary information files. All the data related to this paper are available from the corresponding authors upon reasonable request. A reporting summary for this article is available as a Supplementary information file.

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

## Acknowledgements

This work was supported by the National Natural Science Foundation of China (51873092, 51961160730, 21788102, 31922045, 31771031, and 81701829), the National Key R&D Program of China (Intergovernmental cooperation project, 2017YFE0132200), the National Key Research and Development Program of China (2018YFA0209800), the Research Grants Council of Hong Kong (C6009-17G and A-HKUST605/16), the Innovation and Technology Commission (ITC-CNERC14SC01 and ITCPD/17-9), the Fundamental Research Funds for the Central Universities, Nankai University, China, and the Science and Technology Plan of Shenzhen (JCYJ20160229205601482, JCYJ20170818113348852, and JCYJ20180507183832744).

## Author contributions

J.Q., D.D., X.X., and B.Z.T. conceived and designed the study. J.Q. synthesized and characterized the compounds. J.Q., L.F., and X.Z. performed the NPs preparation and in vitro experiments. L.F. performed the in vivo experiments. H.Z. provided technical assistance with theoretical calculation. J.Q., L.F., L.H., Y.Z., Z.Z., X.D., R.T.K.K., J.W.Y.L., D.D., X.X., and B.Z.T. analyzed the data and participated in the discussion. J.Q., L.F., D.D., X.X., and B.Z.T. contributed to the writing of this paper.

## Competing interests

The authors declare no competing interests.
