## [Peer Review File · Nature Communications]

REVIEWER COMMENTS

Reviewer #1 (Remarks to the Author):

This study appears to contribute to the development of sensitive photoacoustic (PA) probes to detect nitric oxide (NO) related neuropathology. This is accomplished by the development of a sensitive NO probe in PA for in vivo detection. The detection of the probe's good selectivity and sensitivity to NO is also an important finding. However, the study will become more understandable and fluid by explaining the points I have mentioned below and correcting the spelling mistake I have identified. Each point is explained and asked in detail.

1) It is explained that the medium was collected after incubation under the title of Cell culture and stimulation of lipopolysaccharide. In addition, the LPS dose was determined as 100 ng mL⁻¹. At this stage, it is important to determine the lethal dose of LPS by preliminary study.

a) Were these processes taken into account in determining this dose?

b) Were medium dose analyzes performed and evaluated for reagent products?

2) The PL intensity graph in Figure 4c is quite remarkable. However, which unit was given to the figure d under the graph of absorption and nitric oxide concentration?

While preparing this graph, did the ratio of nm and au be taken into account? If so, at least was it graphically observed?

Chemical analysis in the part of facilitating molecular movement in nanoparticles and developing PA biopropes for the detection of NO in vivo has been explained quite well. However, the answers to the following questions are important in this section.

3) As a method in the described Synthesis processes and characterizations section, we clean and sterilize the flask with dry nitrogen once at the angle of control of each stage before vacuuming. Then we proceed to the vacuuming and cleaning stages. Is there any special reason in your method? Because it was washed after 3 stages, not before.

In addition, if magnesium sulfate will be written as MgSO₄, dichloromethane should be written as CH₂Cl₂.

4) As the method, anhydrous 1,4-dioxane was added as 50 mL. Water is very important at this stage. Anhydrous use is actually enough to use 30 mL. We have experienced this. Is there a special reason for increasing the amount?

5) Choosing the 2^{-Delta Delta C (T)} method as the most reliable way to analyze relative changes in gene expression in real-time quantitative PCR experiments is quite the right choice. While applying this method, we can further increase the accuracy of the results with the method of reducing and increasing the dose as a control in our studies. With this method, was a comparison made with the existing data?

6) In some graphs, P value is given as <0.05, <0.001, <0.0001. In general, it is correct to give <0.05, but it would be nice to highlight the <0.0001 findings, which are quite significant and valuable in the study findings, in this section. In other words, it is better to explain the results with a more specific expression rather than a wide expression with a value below <0.05.

7) Microglia and astrocytes are activated during inflammatory neuropathology, and up-regulation of the expression of inducible nitric oxide synthase (iNOS) is accurate information, but has major deficiencies. In the neuroimmunopathological mechanisms of parasitic (*Toxoplasma gondii*) and viral (Pestivirus) encephalitis, both nNOS and eNOS expressions have been shown to increase significantly. Therefore, evaluating iNOS individually will cause an error in the interpretation of

increasing NO concentrations. Because, in the light of the studies carried out in the last five years, neuropathogenesis has been shown to contribute significantly to NO release in neurons and endothelial cells in addition to microglia and astrocytes. These sources need to be checked and reinterpreted.

8) '... due to that higher LPS concentration induces severer encephalitis.' The word Severer should be corrected.

Reviewer #2 (Remarks to the Author):

The manuscript presents a NO-activated PA nanoprobe for in vivo detection of encephalitis. The nanoprobe is constructed by incorporating an AIE core that serves as the PA probe within a DSPE-PEG shell. The AIE can generate a new absorption band in NIR region and enhance PA imaging in response to NO. To show the in vivo detection functionality of the designed nanoprobe the authors used a brain disease model of encephalitis. This is a nicely designed work and well-written. A main question that I have is what makes their PA nanoprobe advantageous over other previously reported ones. A vast number of NO-activated nanoprobe have been reported. The benefits of using this probe should be clearly presented. I have a few suggestions for the authors to consider to further improve the manuscript:

(1) The nanoprobe exhibited different photophysical properties before and after NO exposure. I am wondering the sensitivity of the probe. What is the detection limitation of NO? What is the level of NO at the diseased site? Does it meet the detection limitation? How about the response kinetics?

(2) In vivo PA imaging of encephalitis was shown by intracerebroventricularly injecting the nanoprobe. The intravenous administration route is needed to adopt to show the practical application of the probe.

(3) The nanoprobe are over 100 nm in size, and may be even larger when forming aggregates in salt solutions. I am wondering the possibility to overcome blood brain barrier.

(4) Fig 6 in vivo imaging is based on NO detection. The histological analysis of NO in the tissues should be presented and the correlation of the results to the in vivo PA imaging should be discussed.

(5) The qualities of the figures are very low.

Responses to reviewers' comments for the manuscript titled "Facilitation of molecular motion in nanoparticles and development of turn-on photoacoustic bioprobe for in vivo detection of nitric oxide in encephalitis"

We thank the reviewers for their thoughtful critiques and their recognition that "The detection of the probe's good selectivity and sensitivity to NO is also an important finding" and "This is a nicely designed work and well-written". We appreciate their assessment that "This study appears to contribute to the development of sensitive photoacoustic (PA) probes to detect nitric oxide (NO) related neuropathology. This is accomplished by the development of a sensitive NO probe in PA for in vivo detection" and "The AIE can generate a new absorption band in NIR region and enhance PA imaging in response to NO. To show the in vivo detection functionality of the designed nanoprobe the authors used a brain disease model of encephalitis". We believe that the feedback we received has helped us to substantially increase the impact of our work. Please see below a point-by-point response to the comments:

Responses to the Comments and Suggestions of Reviewer #1

Reviewer #1 (Remarks to the Author):

This study appears to contribute to the development of sensitive photoacoustic (PA) probes to detect nitric oxide (NO) related neuropathology. This is accomplished by the development of a sensitive NO probe in PA for in vivo detection. The detection of the probe's good selectivity and sensitivity to NO is also an important finding. However, the study will become more understandable and fluid by explaining the points I have mentioned below and correcting the spelling mistake I have identified. Each point is explained and asked in detail.

1) It is explained that the medium was collected after incubation under the title of Cell culture and stimulation of lipopolysaccharide. In addition, the LPS dose was determined as 100 ng mL⁻¹. At this stage, it is important to determine the lethal dose of LPS by preliminary study.

a) Were these processes taken into account in determining this dose?

Response: We sincerely thank the reviewer for your careful reading and positive feedback to our work. According to the literatures (Eur. J. Pharmacol. 2010, 648, 110; Neuropharmacology 2011, 61, 592; J. Neurosci. Res. 2007, 85, 1010; Nat. Commun. 2014, 5, 4696), the concentration of 100 ng mL⁻¹ of LPS has been widely used for cell stimulation. We have determined the NO generation ability and biocompatibility of different concentrations of LPS. It shows that the NO generation with LPS concentration higher than 100 ng mL⁻¹ doesn't increase a lot, while the NO stimulated by low concentration of LPS is relatively low. Thus, 100 ng mL⁻¹ of LPS was used for cell stimulation, and the NO concentration generated by 100 ng mL⁻¹ of LPS also matches the NO detection range of the nanoprobe in vitro. Moreover, the MTT results indicate that the LPS doses tested here nearly have no effect on BV-2 cell viability. Accordingly, we believe that the LPS dose of 100 ng mL⁻¹ is suitable for cell stimulation.

Fig. 1 The NO₂⁻ concentrations produced by BV-2 cells after the treatment of different concentrations of LPS. Data are presented as mean ± SD ($n = 3$). **** $P < 0.0001$.

Fig. 2 Cell viability of BV-2 cells with the treatment of different concentrations of LPS. Data are presented as mean ± SD ($n = 4$).

b) Were medium dose analyzes performed and evaluated for reagent products?

Response: The NO concentration in the medium was determined using Griess assay, which is widely used for evaluating NO level. According to previous papers, nitrite is the stable end product of nitric oxide radical, and widely used as the indicator for NO (Free Radic. Biol. Med. 2007, 43,

645; Med. Gas Res. 2019, 9, 192). After BV-2 cells were treated with LPS, NO would be generated and then diffused into the cell medium. Briefly, 50 μL of LPS-treated BV-2 cell medium was incubated with 50 μL of Griess reagent at room temperature for 10 min. The absorbance at 540 nm was measured *via* a microplate reader (SpectraMax i3x). The standard curve was prepared with various known concentrations (ranging from 0.1 to 40 μM) of NaNO_2 . The cell medium was then collected as the NO source for further in vitro experiments. The NO concentration in the medium treated with 100 ng mL^{-1} of LPS was measured to be 22.2 μM . For specific NO concentration, the medium was diluted for certain fold, and the NO concentration of the dilute medium was evaluated by the standard curve. We have added the description in detail in Methods section of the revised manuscript (line 14-23, page 16, line 1-3, page 17).

Fig. 3 The standard curve of Griess assay.

2) The PL intensity graph in Figure 4c is quite remarkable. However, which unit was given to the figure d under the graph of absorption and nitric oxide concentration?

Response: We have provided unit in Fig. 4d and reported the concentration of interfering reactive species used in the figure legend in the revised manuscript.

While preparing this graph, did the ratio of nm and au be taken into account? If so, at least was it graphically observed?

Response: Yes, we have provided the absorption value (au) of OTTAB NPs at different wavelength (nm) of OTTAB NPs with the treatment of different concentrations of NO in the Supplementary information (new Supplementary Fig. 21) in the revised manuscript. The ratio between the absorption intensity at 690 nm and NO concentration (Inset of Fig. 4d) was further analyzed for linear fitting, which suggested good linear relationship ($R^2 = 0.998$) with a slope of 0.021. We have reported it in the revised manuscript (line 15-16, page 9).

Fig. 4 The absorption value of OTTAB NPs at different wavelength upon the treatment of different concentrations of NO (0, 1, 2, 4, 6, 8, 10 μM).

Chemical analysis in the part of facilitating molecular movement in nanoparticles and developing PA biopropes for the detection of NO in vivo has been explained quite well. However, the answers to the following questions are important in this section.

3) As a method in the described Synthesis processes and characterizations section, we clean and sterilize the flask with dry nitrogen once at the angle of control of each stage before vacuuming. Then we proceed to the vacuuming and cleaning stages. Is there any special reason in your method? Because it was washed after 3 stages, not before.

Response: In the revised manuscript, we have provided more detailed description about our cleaning steps and syntheses processes, and the purpose of each step to make them clearer. First, we cleaned and dried the flask for chemical reaction. The flask was first cleaned and blown with nitrogen, and dried in an oven. Before reaction, the flask was vacuumed and heated with a heat gun, then allowed to cool to room temperature in the vacuum state naturally. This process was repeated for three times to remove residual air and water in the flask. Then we performed the chemical reaction, the general procedure was: the solid chemicals were firstly added into the flask, which was vacuumed and purged with dry nitrogen three times to remove the air in the reaction system. Then solvents or liquid reagents were added into the flask, and chemical reaction could be carried out in nitrogen atmosphere. This method has also been used by other groups from the literatures (Angew. Chem. Int. Ed. 2007, 46, 5359; Angew. Chem. Int. Ed. 2020, 59, 9288). We have added the detailed description in the revised Supplementary information (line 7-9, page 2, line 1, page 3).

In addition, if magnesium sulfate will be written as MgSO4, dichloromethane should be written as CH2Cl2.

Response: We thank for the suggestion from the reviewer, and we have written dichloromethane as CH₂Cl₂ in the Supplementary information of the revised manuscript.

4) As the method, anhydrous 1,4-dioxane was added as 50 mL. Water is very important at this stage. Anhydrous use is actually enough to use 30 mL. We have experienced this. Is there a special reason for increasing the amount?

Response: The reaction mechanism of Miyaura-Ishiyama borylation is shown in the following scheme (J. Am. Chem. Soc. 2002, 124, 8001; Org. Lett. 2011, 13, 1366; Chem. Rec. 2014, 3, 271). We used anhydrous 1,4-dioxane as the solvent in this work, which is widely used for this reaction in the literatures (Angew. Chem. Int. Ed. 2007, 46, 5359; Synlett 2018, 29, 1055). To ensure the raw materials can be dissolved sufficiently, we used 50 mL of anhydrous 1,4-dioxane. According to the reviewer's suggestion, we have compared the reaction yields of using 50 mL and 30 mL of anhydrous 1,4-dioxane as solvent. It shows that 30 mL of anhydrous 1,4-dioxane gives a little higher yield (87% vs 90%). Thanks for the reviewer's great suggestion and we will pay attention to the solvent volume in the future. We have used 30 mL of anhydrous 1,4-dioxane as the solvent in the revised manuscript.

Scheme 1. Reaction mechanism of Miyaura-Ishiyama borylation.

5) Choosing the 2 (-Delta Delta C (T)) method as the most reliable way to analyze relative changes

in gene expression in real-time quantitative PCR experiments is quite the right choice. While applying this method, we can further increase the accuracy of the results with the method of reducing and increasing the dose as a control in our studies. With this method, was a comparison made with the existing data?

Response: We thank for the suggestion from the reviewer. We suppose that the reviewer advises us to study the mRNA levels of iNOS stimulated with different LPS concentrations to increase the accuracy. We have measured the mRNA levels of iNOS with more different LPS concentrations. The results are shown in the following, which provide higher accuracy about the relationship between the concentration of LPS and the concentration of iNOS. iNOS can catalyze the oxidation of *L*-arginine to *L*-citrulline to generate NO. Therefore, the expression of iNOS reflects the NO level directly, which can be determined by noninvasive PA imaging. As a result, the relationship of “encephalitis severity-iNOS level-NO concentration-PA intensity” has been established in this work. We have added the results in new Fig. 6c.

Fig. 5 Relative inducible nitric oxide synthase (iNOS) mRNA level at the disease site versus different concentrations of LPS injected into mouse brain. Data are presented as mean \pm SD ($n = 3$ mice). * $P < 0.05$, *** $P < 0.001$, **** $P < 0.0001$.

6) In some graphs, P value is given as <0.05, <0.001, <0.0001. In general, it is correct to give <0.05, but it would be nice to highlight the <0.0001 findings, which are quite significant and valuable in the study findings, in this section. In other words, it is better to explain the results with a more specific expression rather than a wide expression with a value below <0.05.

Response: We thank for the suggestion from the reviewer. We have listed all accurate *P* values in the Supplementary information in the revised manuscript (new Supplementary Tables 1-4).

*7) Microglia and astrocytes are activated during inflammatory neuropathology, and up-regulation of the expression of inducible nitric oxide synthase (iNOS) is accurate information, but has major deficiencies. In the neuroimmunopathological mechanisms of parasitic (*Toxoplasma gondii*) and viral (Pestivirus) encephalitis, both nNOS and eNOS expressions have been shown to increase significantly. Therefore, evaluating iNOS individually will cause an error in the interpretation of increasing NO concentrations. Because, in the light of the studies carried out in the last five years, neuropathogenesis has been shown to contribute significantly to NO release in neurons and endothelial cells in addition to microglia and astrocytes. These sources need to be checked and reinterpreted.*

Response: We truly agree with the reviewer's comments. The three isoforms of NOS (eNOS, nNOS and iNOS) are all very important for the generation of NO (Eur. Heart J. 2012, 33, 829; Exp. Parasitol. 2015, 156, 104). In different kinds of encephalitis, one or two or all three NOS isoforms may be the main contributor to NO production. In the LPS-induced encephalitis model, we have measured and compared the mRNA levels of eNOS, nNOS and iNOS. The expression of nNOS and eNOS in inflamed brain is a little higher than that of normal one, yet much lower than

the expression of iNOS. Therefore, the up-regulation of iNOS is considered as the main contribution to NO production in this work. Moreover, we chose three kinds of cells including bEnd 3 (endothelial cell), N2A (neuron) and BV-2 cell (microglia) to investigate the NO production in different cell types after LPS stimulation. The NO generated from LPS-activated BV-2 cells is much higher than that from the other two cell lines. Taken together, we think the iNOS expression in microglia and astrocytes might be the major contributor to NO production in our animal model, so we use the expression of iNOS as the representative of NOS. But we totally agree with the reviewer that the expression of eNOS, nNOS and iNOS in several cell types (including endothelial cells, neurons, microglia and astrocytes) might contribute to the generation of NO in different encephalitis models. We have added these points in our revised manuscript (new Supplementary Fig. 25,26, line 12-19, page 12).

Fig. 6 The NO_2^- concentrations generated from bEnd 3, N2A and BV-2 cells after the same LPS stimulation. The bEnd 3 (endothelial cell), N2A (neuron) and BV-2 cell (microglia) were used to investigate the NO production after LPS stimulation. The NO generation ability from BV-2 cell is very high, and the other two can be negligible. Data are presented as mean \pm SD ($n = 3$). ** $P < 0.01$.

Fig. 7 Relative mRNA levels of eNOS, nNOS and iNOS at the disease site in the LPS-induced encephalitis model. Data are presented as mean \pm SD ($n = 3$ mice). ** $P < 0.01$.

8) *'... due to that higher LPS concentration induces severer encephalitis.'* The word *Severer* should be corrected.

Response: We are sorry about it, and we have changed the word “severer” to “more severe” in the revised manuscript (line 7, page 13).

Responses to the Comments and Suggestions of Reviewer #2

Reviewer #2 (Remarks to the Author):

The manuscript presents a NO-activated PA nanoprobe for in vivo detection of encephalitis. The nanoprobe is constructed by incorporating an AIE core that serves as the PA probe within a DSPE-PEG shell. The AIE can generate a new absorption band in NIR region and enhance PA imaging in response to NO. To show the in vivo detection functionality of the designed nanoprobe the authors used a brain disease model of encephalitis. This is a nicely designed work and well-written. A main question that I have is what makes their PA nanoprobe advantageous over other previously reported ones. A vast number of NO-activated nanoprobe have been reported. The benefits of using this probe should be clearly presented. I have a few suggestions for the authors to consider to further improve the manuscript:

Response: We sincerely thank the reviewer for your careful reading and positive feedback to our work. Yes, some NO-activated nanoprobes have been reported previously, most of which are based on fluorescence imaging. Fluorescence imaging has high sensitivity, yet it faces the drawback of shallow penetration depth, which restricts its use to the cellular level and skin surfaces. The new NO probes with high penetration and spatiotemporal resolution should be developed for in vivo applications. In this work, we report a sensitive high-performance PA probe for NO detection and related encephalitis diagnosis. The advantages of our probe are: 1. The probe could react with NO to yield a relatively planar structure with strong intramolecular donor-acceptor interaction, and thus a new strong absorption band in near-infrared spectral region, enabling turn-on PA signal. PA technique could combine the benefits of optical resolution and acoustic depth of penetration, which enables deep-tissue imaging capacity with high spatial resolution and real-time monitoring. The turn-on property enables a high signal-to-noise ratio, but NO probe with turn-on PA signature has

been seldom reported previously. The turn-on bioprobe in this work would be beneficial for in vivo diagnosis of complicated and deeply located diseases, like NO-related encephalitis. 2. The intense molecular motion in nanoparticles could facilitate the active intramolecular motion and thus boost PA conversion. By making full use of the intramolecular motion, the absorbed light could convert into acoustic signal maximally, and a highly sensitive NO probe is obtained. The new design strategy of fully utilizing the molecular motion in NPs would shed light on the exploitation of more sensitive bioprobes. 3. The probe has been used for noninvasive in vivo PA imaging of NO and real-time detecting encephalitis and differentiating its severity, which represents the first reported example of NO-activated probe for in vivo PA imaging of encephalitis in situ. The high scattering of skull restricts the noninvasive imaging of brain disease (e.g., encephalitis) with fluorescence technique, while the PA probe developed in this work can overcome this limitation and realize high-performance disease diagnosis in situ. 4. The probe also possesses the merits of high selectivity and sensitivity and good biocompatibility, rendering great promise for real applications. We have made more discussion about the advantages of the NO-activated PA nanoprobe in the discussion section of revised manuscript (line 8-9, 11-12, 13-16, 19-21, page 14).

(1) The nanoprobe exhibited different photophysical properties before and after NO exposure. I am wondering the sensitivity of the probe. What is the detection limitation of NO? What is the level of NO at the diseased site? Does it meet the detection limitation? How about the response kinetics?

Response: OTTAB NPs showed a good linear correlation ($R^2 = 0.998$) towards NO concentration, and the detection limit ($3\sigma/\text{slope}$) of OTTAB NPs for NO was calculated to be 377 nM. We have reported the detection limit in the revised manuscript (line 15-16, page 9). The NO concentration

at the disease site was measured to be 6.3 μM when 500 $\mu\text{g}/\text{kg}$ of LPS was administered, which is much higher than the detection limit of our probe. The reaction kinetics of OTTAB NPs towards NO is presented in the following. We have provided the response kinetics in the revised manuscript (new Supplementary Fig. 20).

Fig. 1 The NO_2^- concentration at the disease site versus the concentration of LPS injected into mouse brain. Data are presented as mean \pm SD ($n = 3$ mice). * $P < 0.05$, ** $P < 0.01$, *** $P < 0.001$, **** $P < 0.0001$.

Fig. 2 Absorption of OTTAB NPs at 690 nm upon reaction with NO for different time.

(2) In vivo PA imaging of encephalitis was shown by intracerebroventricularly injecting the nanoprobe. The intravenous administration route is needed to adopt to show the practical application of the probe.

Response: According to the reviewer's suggestion, in vivo PA imaging of encephalitis after intravenous administration of the probe has been demonstrated in the revised manuscript. We totally agree with the reviewer that intravenous administration is more suitable for practical application. It is very hard for normal nanomaterials to diagnose and treat brain diseases because brain is protected by the blood-brain barrier (BBB), which gives tight control over the passage of substances moving from blood to the tissues of the central nervous system. Currently, several strategies have been developed to enhance the penetration of NPs across BBB, including incorporating certain functional targeting ligands or special cell membrane on NPs (Nat. Nanotechnol. 2017, 12, 692; Proc. Natl. Acad. Sci. U.S.A. 2020, 117, 19141). For example, neutrophils (NEs) have been reported to possess an innate ability to penetrate the BBB, which has been utilized to deliver NPs into brain (Nat. Nanotechnol. 2017, 12, 692; Nat. Nanotechnol. 2018, 13, 1182; ACS Nano 2015, 9, 11800). In this work, NEs were employed as a carrier to deliver our probe to brain after intravenous injection. The NE-delivered OTTAB nanoprobe (OTTAB@NEs) were obtained by incubating NEs with OTTAB NPs, after which OTTAB NPs could be taken up efficiently by NEs due to the strong phagocytic capabilities of NEs. The obtained OTTAB@NEs showed similar NO-responsive ability as OTTAB NPs (Fig 3). The PA imaging results of the intravenously injected OTTAB@NEs at disease site are shown in Fig. 4, which suggest that our probe can be used for in vivo imaging of encephalitis after intravenous administration. We have

added the results as new Supplementary Fig. 29,30 and discussed them in the revised manuscript (line 14-22, page 13, line 1-2, page 14).

Fig. 3 PA amplitudes of OTTAB@NEs without and with the treatment of NO. Data are presented as mean \pm SD ($n = 3$). *** $P < 0.001$.

Fig. 4 Representative in vivo noninvasive PA images and the corresponding PA intensity of LPS-induced encephalitis after intravenous administration of NEs and OTTAB@NEs. Scale bar: 1 mm. Data are presented as mean \pm SD ($n = 3$ mice). *** $P < 0.001$.

(3) The nanoprobe are over 100 nm in size, and may be even larger when forming aggregates in salt solutions. I am wondering the possibility to overcome blood brain barrier.

Response: Yes, size is one of the important factors that influence the penetration of BBB. In fact, it is also very hard for small-sized nanoparticles and small molecule drugs without any modification to pass through BBB. However, several efficient strategies have been developed to increase the penetration of NPs (including NPs > 100 nm) across BBB. For example, NPs can be functionalized with certain surface ligands that interact with receptors expressed on BBB to enhance BBB penetration (Nat. Nanotechnol. 2017, 12, 692; Nat. Commun. 2018, 9, 1991; Sci. Adv. 2020, 6, eabc7031). Additionally, several specific types of cells have been recognized as bioinspired and powerful platforms to deliver NPs into brain (Nat. Nanotechnol. 2017, 12, 692; Sci. Adv. 2020, 6, eabc7031). For example, neutrophils (NEs) possess the native ability of crossing BBB and infiltrating the inflammation site (Nat. Nanotechnol. 2017, 12, 692; Nat. Nanotechnol. 2018, 13, 1182; ACS Nano 2015, 9, 11800). In the revised manuscript, NEs were utilized as the carriers to enhance the BBB crossing ability of OTTAB nanoprobe. The OTTAB NPs was incubated with neutrophils (NEs) to obtain OTTAB@NEs for penetrating BBB. The results showed that OTTAB@NEs maintained the NO-responsive ability and could image brain inflammation efficiently after intravenous administration of OTTAB@NEs. We have added these results in the revised manuscript. In addition, we have also studied the stability of OTTAB NPs. The results show that the nanoprobe is stable and the size nearly doesn't change in different conditions. We have added the stability study as new Supplementary Fig. 18 and reported it in the revised manuscript (line 11-13, page 8).

Fig. 5 Average diameters of OTTAB NPs in different conditions: (a) Dulbecco's Modified Eagle's Medium (DMEM) and (b) different pH environments. Data are presented as mean \pm SD ($n = 3$).

(4) Fig 6 in vivo imaging is based on NO detection. The histological analysis of NO in the tissues should be presented and the correlation of the results to the in vivo PA imaging should be discussed.

Response: The detection of NO radicals in biological tissues is particularly difficult due to the short lifetime in tissues. Therefore, some indirect methods have been developed to determine NO concentration. For example, the Griess assay is widely used for evaluating NO level by measuring nitrite, the stable end product of NO radical (Free Radic. Biol. Med. 2007, 43, 645; Med. Gas Res. 2019, 9, 192). Therefore, Griess reagent was used to measure the NO concentration in mouse brain in this work. As shown in the following, the NO concentrations in mouse brain are elevated drastically with the doses of LPS used for treatment, which is consistent well with the increased in vivo PA signal from the nanoprobe. Additionally, to further show the correlation of PA imaging with NO generation, we also studied the expression of iNOS (inos nitric oxide synthases), which is a key enzyme to catalyze the oxidation of *L*-arginine to *L*-citrulline to generate NO. As shown in the previous PCR data (Fig 6c) and the new data of immunofluorescent staining of iNOS in

brain tissue (Supplementary Fig. 27), the expression level of iNOS in inflamed brain tissue is much higher than normal brain. The increased expression of iNOS in inflamed site correlates with the elevated NO concentration, which was also determined by the enhanced in vivo PA signal. As a result, the relationship of “encephalitis severity-NOS level-NO concentration-PA intensity” has been established in this work. We have added the results as new Supplementary Fig. 27,28 and discussed them in the revised manuscript (line 21-23, page 12, line 1-4, page 13).

Fig. 6 The NO₂⁻ concentration at the disease site versus the concentration of LPS injected into mouse brain. Data are presented as mean ± SD (*n* = 3 mice). **P* < 0.05, ***P* < 0.01, ****P* < 0.001, *****P* < 0.0001.

Fig. 7 Immunofluorescent staining of iNOS in (a) normal and (b) inflamed brains. Nuclei were stained with DAPI (blue signal), and iNOS was stained with the anti-iNOS primary antibody followed by FITC-labeled secondary antibody (green signal). Scale bar: 250 μ m.

(5) The qualities of the figures are very low.

Response: We are sorry for the low-quality figures, and we have provided high-quality figures in the revised manuscript.

REVIEWERS' COMMENTS

Reviewer #1 (Remarks to the Author):

Dear Authors;

The corrections made are sufficient and good. I do not see any missing.

Reviewer #1 comments on reviewer #2 concerns:

The second referee first asked to explain the advantages of the developed probe in the article in detail. The authors explained its advantages in detail. I also have things that I want to add as an advantage. We commonly use NO activated nanoprobes in our research at the fluorescence imaging stage. If we evaluate it in terms of sensitivity at this stage, we can evaluate it as a sensitive but prolonged process. I see that it is important from the findings that this probe has high selectivity because it is sensitive.

Another issue that the referee is curious about is the biophysical detection features of the probe. The authors also explained this part and made additions to the article. However, in the non-revised article, I see that they refer to these sections with statistical data and tables.

Another issue that the referee is curious about is the practical application of the probe. Studies performed intracerebroventricular include the highest level and sensitivity. The first step in the central nervous system, its powerful control mechanism is the blood-brain barrier. Practices that directly and quickly affect the blood brain barrier make it easier to think at a lower level. Therefore, the data to be obtained at this stage will precisely lead intravenous applications. The authors explain this part with reference support. We have experienced this in many experimental models on this subject.

The corrections were made and explained, and I reviewed them one by one. The histopathological photos added in the revision pdf should be added to the main article together with the prepared tables. Therefore, it will be beautiful and understandable with such addition.

We sincerely thank the reviewer for his/her precious time and recognition of our work. Below are our point-to-point responses to the reviewer's comments.

Responses to the comments and suggestions of Reviewer #1

Reviewer #1 (Remarks to the Author):

Dear Authors;

The corrections made are sufficient and good. I do not see any missing.

Response: We sincerely thank the reviewer for your careful reading and supporting publication of our study.

Responses to the comments and suggestions of Reviewer #1's comments on Reviewer #2's concern

Reviewer #1 comments on reviewer #2 concerns:

The second referee first asked to explain the advantages of the developed probe in the article in detail. The authors explained its advantages in detail. I also have things that I want to add as an advantage. We commonly use NO activated nanoprobes in our research at the fluorescence imaging stage. If we evaluate it in terms of sensitivity at this stage, we can evaluate it as a sensitive but prolonged process. I see that it is important from the findings that this probe has high selectivity because it is sensitive.

Response: We sincerely thank reviewer 1 for your precious time and careful reading. We have added description about the advantage mentioned by the reviewer in the revised manuscript as "The probe not only exhibits good selectivity and quantitation towards NO, but also possesses high sensitivity, which enable excellent in vitro PA signal output" (line 8-10, page 5).

Another issue that the referee is curious about is the biophysical detection features of the probe. The authors also explained this part and made additions to the article. However, in the non-revised article, I see that they refer to these sections with statistical data and tables.

Response: Yes, the added detection properties of the probe makes the article clearer and more understandable.

Another issue that the referee is curious about is the practical application of the probe. Studies performed intracerebroventricular include the highest level and sensitivity. The first step in the central nervous system, its powerful control mechanism is the blood-brain barrier. Practices that directly and quickly affect the blood brain barrier make it easier to think at a lower level. Therefore, the data to be obtained at this stage will precisely lead intravenous applications. The authors explain this part with reference support. We have experienced this in many experimental models on this subject.

Response: We agree with the reviewer's comments that blood-brain barrier (BBB) is the most powerful control of the central nervous system. The utilization of neutrophils (NEs) as the carrier could penetrate BBB efficiently, and the experimental results obtained in this work suggests promising applications by intravenous injection of the probe.

The corrections were made and explained, and I reviewed them one by one. The histopathological photos added in the revision pdf should be added to the main article together with the prepared tables. Therefore, it will be beautiful and understandable with such addition.

Response: We thank for the reviewer's suggestion. We have added the histopathological results to the main article (Fig. 6f). And we have also provided the p values in the figure legend (Fig. 5c,6) of the revised manuscript.